# The Pathogenic Role of PI3K/AKT Pathway in Cancer Onset and Drug Resistance: An Updated Review

**DOI:** 10.3390/cancers13163949

**Published:** 2021-08-05

**Authors:** Federica Rascio, Federica Spadaccino, Maria Teresa Rocchetti, Giuseppe Castellano, Giovanni Stallone, Giuseppe Stefano Netti, Elena Ranieri

**Affiliations:** 1Nephrology Dialysis and Transplantation Unit, Advanced Research Center on Kidney Aging (A.R.K.A.), Department of Medical and Surgical Sciences, University of Foggia, 71122 Foggia, Italy; giuseppe.castellano@unifg.it (G.C.); giovanni.stallone@unifg.it (G.S.); 2Clinical Pathology Unit, Advanced Research Center on Kidney Aging (A.R.K.A.), Department of Medical and Surgical Sciences, University of Foggia, 71122 Foggia, Italy; federica.spadaccino@unifg.it (F.S.); giuseppestefano.netti@unifg.it (G.S.N.); elena.ranieri@unifg.it (E.R.); 3Cell Biology Unit, Department of Clinical and Experimental Medicine, University of Foggia, 71122 Foggia, Italy; mariateresa.rocchetti@unifg.it

**Keywords:** mTOR, Akt/PI3K pathway, tumour, drug resistance, regulatory mechanism, miRNA

## Abstract

**Simple Summary:**

Drug resistance remains one of the major problems in cancer therapy and is responsible for up to 90% of cancer-related deaths. It exists across all types of cancer and treatment; thus, determining how to overcome this problem is a goal that involves understanding biological mechanisms and also includes clinical trials to test new therapeutic strategies. In this review, we will highlight the emerging role of the PI3K/Akt pathway in drug resistance by discussing recent findings of a multi-level deregulation. Combinational and personalized therapies, which should take this pathway into consideration, might provide better treatment strategies and improved efficacy for fighting drug resistance in cancer.

**Abstract:**

The PI3K/AKT pathway is one of the most frequently over-activated intracellular pathways in several human cancers. This pathway, acting on different downstream target proteins, contributes to the carcinogenesis, proliferation, invasion, and metastasis of tumour cells. A multi-level impairment, involving mutation and genetic alteration, aberrant regulation of miRNAs sequences, and abnormal phosphorylation of cascade factors, has been found in multiple cancer types. The deregulation of this pathway counteracts common therapeutic strategies and contributes to multidrug resistance. In this review, we underline the involvement of this pathway in patho-physiological cell survival mechanisms, emphasizing its key role in the development of drug resistance. We also provide an overview of the potential inhibition strategies currently available.

## 1. Introduction

In the last few years, the lack of success of therapeutic approaches for cancer, mainly caused by intrinsic or acquired drug resistance, underlines the importance of the investigation of the underlying molecular mechanisms and potential therapeutic targets associated with the progress of tumours [1]. The phosphatidylinositol-3-kinase PI3K/Akt/mammalian target of the rapamycin (mTOR) pathway is one of the most frequently activated intracellular pathways, which is commonly involved as a balancer for human cancer [2]. This pathway is involved in the regulation of multiple cellular physiological processes, particularly cell proliferation, cell cycle, and apoptosis [3]. Several studies have indicated that the occurrence and development of different type of cancers, especially solid ones, depend at least partly on the PI3K/Akt pathway’s deregulation. Thus, enormous effort has been made to find new drugs targeting PI3K/AKT signalling, downstream or upstream of the pathway [4,5,6].

Undoubtedly, the oncogenic variations of the PI3K pathway may be involved in the resistance to specific antineoplastic therapies, including chemotherapies [7]. The abnormal activation of the PI3K/AKT pathway, along with up- or downstream targets transduction, plays a crucial role as an important signalling pathway in charge of drug resistance in many types of neoplasia [8].

The principal post-transcriptional regulators of human genes are microRNAs (miRNAs), small non-coding RNAs which act as direct or indirect modulators of cellular processes, including cell proliferation, differentiation, apoptosis, and invasion [9]. Indeed, even the PI3K/Akt pathway, which plays a key role in carcinogenic processes, is heavily regulated by miRNAs [10].

In this review, we have summed up the aberrant activation of the PI3K/Aky pathway as a link between the multidrug resistance (MDR) and dysfunction of miRNAs that act by dysregulating this signalling pathway. Furthermore, the role of this pathway and miRNAs in target therapies against cancers are discussed.

## 2. Structure, Activation, and Function of the PI3K/Akt Cascade

PI3K/Akt signalling is an important pathway formed by two parts: phosphatidylinositol-3-kinase (PI3K) and a serine/threonine protein kinase B (PKB) [11].

The phosphoinositide 3-kinases (PI3Ks) are an important family of lipid enzymes which are triggered by phosphorylation of the 3-idroxyl group of phosphatidylinositol on the plasma membrane [2]. There are four different classes of PI3K according to their affinity for lipid substrates and specific structures controlling different aspects of cell biology. Among them, class I PI3K, which includes heterodimeric enzymes composed by a regulatory subunit p85 and a catalytic subunit p110, is the most thoroughly studied isoform in the cancer scenario [12].

Mutations that occur in oncogenes, such as PI3KCA gene encoding the subunit p110, and in tumour suppressor genes, such as phosphatase and tensin homolog (PTEN), are the major mutations responsible for the dysregulation of this pathway in human tumours [13,14].

Generally speaking, PI3K is activated by extracellular signals, such as growth factors, cytokines, and hormones.

In particular, the binding of epidermal growth factor receptor (EGFR), fibroblast growth factor receptor (FGFR), and insulin-like growth factor I receptor (IGF-IR) to the N-terminal extracellular domain of the transmembrane receptor tyrosine kinase (RTK) causes the auto-phosphorylation of tyrosine residues in the cytoplasmic region of the receptor. This event is responsible for the recruitment and activation of PI3K [15,16].

In addition, PI3K may be directly or indirectly triggered by small Ras-related GTPases, well known for their involvement in carcinogenesis.

RAS is one of the major oncogenes in human cancers, and PI3K is believed to be its main effector. Interacting with different isoforms of class I PI3Ks, RAS binds to and activates p110γ catalytic subunit, leading to an increase in PI3K activity. PI3K/RAS interaction has been shown to be indispensable for maintaining cancer cell survival and proliferation [17].

Class I PI3Ks, a protein heterodimer composed of two different proteins, a regulatory and a catalytic subunit, are also divided into two categories, IA and IB subsets, on sequence similarity. Particularly, class IA PI3K binds, with its regulatory subunit, the tyrosine receptors on cell membrane stimulated by ligands and determines the activation of the catalytic subunit that converts PIP2 to the second messenger PIP3 [18,19,20].

The most significant downstream effector of PI3K is the serine/threonine kinase Akt, which plays a crucial role in the signal transduction of the entire signalling cascade [21]. Akt1, Akt2, and Akt3 are three different isoforms of Akt in mammals, which are activated in an isoform-specific manner by many signals, including cytokine receptors, integrins, and B and T cell receptors [22,23]. Akt activation depends on the phosphorylation of both the internal catalytic domain and regulatory domain, but its stabilization requires the phosphorylation of other multiples sites. Subsequently to activation, Akt transfers from the cell membrane to other cell regions to phosphorylate and initiate downstream effector molecules [24].

There are several Akt target proteins; most of are transcription factors which may induce, if mutated, alterations in cancer cells function and metabolism. Among these, glycogen synthase kinase 3 (GSK3) is inhibited in response to growth factors via AKT-mediated phosphorylation, leading to ubiquitylation and proteasomal degradation of its targets involved in cellular metabolism [25].

AKT phosphorylation induces cancer development and progression by inhibiting members of the FOXO family of transcription factors (FOXO1, FOXO3A, FOXO4), which are physiologically involved in the suppression of growth and proliferation [26].

The main target of PI3K/Akt is the serine/threonine kinase mTOR. mTOR includes mTOR complex 1 (mTORC1), whose components are mTOR, Raptor, and mLST8 subunits, and mTOR complex 2 (mTORC2), which consists of mTOR, Rictor, Sin1, and mLST1 [27,28]. Akt induces the phosphorylation and inhibition of the tuberous sclerosis complex 2 (TSC2) protein which, in turn, inhibits Ras homologue enriched in brain (RHEB) and causes mTORC1 activation. Thus, the mTORC1 complex is part of the PI3K/Akt/mTOR pathway as a downstream molecule which promotes cell growth, protein synthesis, and energy storage [28].

This complex is stimulated by the presence of nutrients and depends on the energetic status for its activity. On the contrary, in conditions of starvation, mTORC1 is inhibited and may contribute to an increased apoptosis [29]. Undoubtedly, when mTORC1 is inhibited, autophagy is induced by AMPK, which causes the creation of energy for the survival of cells [30].

The phosphorylation of mTOR is considered a specific marker for tumour progression and, in turn, mediates the activation of downstream target factors, such as S6K and 4E-BP1. In particular, phosphorylated 4E-BP1 (p4E-BP1) is associated with a greater risk of progression in many neoplasms and correlates with aggressive pathologic grade and poor prognosis [31,32,33,34].

## 3. Altered PI3K/Akt Signalling Pathway: Its Pathogenic Role in Human Cancer

The PI3K/Akt/mTOR pathway cascade is frequently disrupted, more often over-activated, in different human malignant cases [5,6,7,8,9,10,11,12,13,14,15,16,17,18,19,20,21,22,23,24,25,26,27,28,29,30,31,32,33,34,35]. Indeed, more and more evidence has demonstrated that the hyper-activation of the PI3K/Akt pathway contributes to the carcinogenesis, proliferation, invasion, metastasis, and drug resistance of tumour cells [5]. Among the various genetic factors involved in the molecular mechanisms of human cancer, such as P53, NFKB, STAT3, or Myc, the PI3K/Akt axis receives major attention, justified by its central role in multiple cellular processes.

The PI3K/Akt/mTOR pathway is not only involved in the regulation of the proliferation and apoptosis of cancer cells but also promotes normal and tumour angiogenesis [36].

Genetic alterations have been identified in every level of the PI3K/Akt signalling cascade. The phosphatidylinositol-4,5-bisphosphate 3-kinase, catalytic subunit alpha (PIK3CA), also called p110 alpha protein, is a class I PI3K catalytic subunit that, at first, was considered to be highly involved in ovarian cancers, playing a role as an oncogene [37], however, the somatic mutations of PIK3CA have also been detected in colorectal, glioblastoma, gastric, breast, lung, and kidney cancers [38,39,40]. Similarly, recurring mutations of PIK3R1 gene coding, in detail for the p85 regulatory subunit, have also been detected in endometrial, glioblastoma, uterine carcinosarcoma, and bladder urothelial carcinoma [41].

The hyper-activation of the PI3K/Akt pathway, caused by mutations of PI3K family genes, determines the poor prognosis in cancers of the brain and of the central nervous system [42,43,44]; the knockdown of the same genes significantly inhibits the invasion of tumours through the hypo-activation of Akt [45]. Aberrant activation of PI3K/Akt signalling in the cancer of the endocrine system is caused by the same genetic mutations.

The second most mutated or deleted tumour suppressor after p53, across all cancers, is the phosphatase and tensin homolog deleted on chromosome 10 (PTEN), the lipid phosphatase that counteracts the PI3K pathway. It is well known that PTEN negatively regulates PI3K/AKT acting as a tumour suppressor by dephosphorylating PIP3, and its function loss leads to an increase in PI3K/AKT pathway activity. As a consequence, PTEN alterations, mutations, and deletions can induce tumorigenesis and other diseases [46,47].

In terms of Akt, the main member of the PI3K/Akt pathway, regulating the effector targets by phosphorylation, many alterations have been suggested in cancer scenario. First of all, the over-expression of Akt2 isoform has already been suggested not only in ovarian cancers but also in several other solid organ neoplasias [48,49]. In addition, more evidence has indicated that Akt2 and Akt3 isoforms are more frequently amplified than Akt1 in cancer disorders [50].

In addition, lipid and protein phosphatases controlling Akt activity are frequently downregulated in human cancer determining the overexpression of phosphorylated Akt, such as inositol polyphosphate 4-phosphatase (INPP4), which catalyses the dephosphorylation of PI (3,4)P2 to PI(3)P at the plasma membrane and on endomembranes, leading, if it is reduced, to the subsequent increase of PI3K/Akt pathway activity [8,51]. Activated Akt produces eNOS allocation in the vascular endothelium, which culminates in angiogenesis and improves the transcriptional activity of NF-kappa B, thus promoting and supporting the proliferation of tumours [52,53,54]. Consequently, when growth factors boost and activate PI3K, its effector Akt further activates downstream molecules and regulates the proliferation, invasion, metastasis, and angiogenesis of cancer cells.

At the layer downstream of Akt, the most common deregulated mechanisms involve the effector target mTOR, responsible for tumour growth progression, as in endometrial malignancies [55,56,57]. On the other hand, it has been demonstrated that the inhibition of mTOR drastically reduces the progression of endometrial cancer [58].

In ovarian cancer, the PI3K/Akt signalling pathway is dysregulated due to mutations in PI3K, PI3KCA, and PTEN deletion, in the regulatory and catalytic domains, or due to modifications of downstream targets of PI3K [59,60,61,62]. Thus, it is clear that the dysregulation of the whole pathway, at multiple layers, rather than individual genes is required to induce tumour progression and altered phenotype in gynaecological cancer disorders [59]. Nevertheless, it has been clarified that PI3KCA mutations exclude alterations in other components of the pathway, with the exception of endometrial or bowel cancers, where a multi-level impairment of the entire pathway is frequently involved [63,64]. In renal clear cell carcinoma (RCC), cancer classified as highly resistant to traditional treatments such as chemotherapies or radiotherapies, the discovery of new diagnostic or prognostic biomarkers has been potentially fruitful and, in part, explored [65,66]. With regard to the PI3K/Akt pathway, although the mutations affecting its components are limited, the downstream effect is a constitutive activation of the cascade [67]. Genetic mutations in its members are frequently observed in RCC, which exacerbate the hyper-activation of the PI3K/Akt/mTOR signalling pathway. However, in patients with clear cell RCC, the mRNA level of the expression of the principal molecules involved in the pathway demonstrate no significant differences if compared to normal tissues, suggesting that the activation of the cascade may not depend on their transcriptional regulation [59]. This evidence indicates that the pathway activity in RCC not only depends on the mRNA and protein expression levels, but it also relies on protein activation through phosphorylation, as demonstrated by the function of phosphor-mTOR and phosphor-Akt. In conditions of deprivation of glucose, multiple cell lines show a unique form of Akt phosphorylation on Thr308, which represents a central Akt activation mechanism in RCC (Figure 1) [67,68].

## 4. miRNAs’ Regulation on the PI3K/Akt Pathway

Cancer is the result of an intricate interaction of genetic and environmental factors. Recent studies have evidenced the encoding of microRNAs (miRNAs) as participants in the pathogenesis of cancer, emerging as a new class of important regulators of the PI3K/AKT pathway [69,70]. These small-size RNA molecules regulate the expression of target genes by binding with their 3′ untranslated regions or, alternatively, with 5′ untranslated regions [71,72,73]. The altered expression of miRNAs in cancer cells works by de-regulating the expression of genes related to apoptosis or autophagy that play a central role in the mechanisms of action of drug resistance [74]. As a consequence, it is well known that miRNAs are potential modulators in the initiation and progression of various disorders. Numerous studies investigating the regulation of miRNAs on the PI3K/Akt pathway are summarised in Table 1. For instance, in RCC cancer, miR-182-5p has been shown to be a negative regulator of Akt, so that its upregulation results in Akt inhibition and a subsequent slowing down of cancer cell proliferation [75]. On the other hand, miR-122 has been demonstrated to be a positive modulator of the pathway [76]. Finally, miR-205-5p as well as miR-182-5p functions as a negative regulator by targeting VEGFA and the PI3K/Akt signalling pathway [77].

In gastric cancer, dysregulated miRNAs have been identified as being partly responsible for malignant processes, including drug resistance, depending on the aberrantly activated PI3K pathway, normally controlled by miRNAs [78]. Indeed, in the gastric system, the constitutive activation of this pathway that promotes cancer is due to epigenetic or genetic alterations regarding the components of this cascade [79,80]. However, in addition to genetic modifications, the onset of cancer depends on the oncogenic effects of dysregulated miRNAs and on the altered activation of their mediators, the PTEN/PI3K/Akt pathway included [81,82]. Interestingly, it was observed that miR-196b promotes gastric cancer by stimulating the pathway activation, while the expression of tumour-suppressor miRNAs (miR137, miR-34, miR203, miR15a miR16, and miR375) exert their effects by targeting Akt2 [83,84,85,86,87,88,89].

Other proteins of the pathway, including PIK3CB and mTOR, are inhibited by miR125b-2, miR-451a, and miR101-2 [90]. Based on these forms of evidence, the potential applications of miRNAs for drugs, including miRNA mimics and inhibitors, in combination with PI3K/AKT pathway inhibitor has been well documented in the treatment of gastric cancer [9]. 

In bladder cancer, the expression of several oncomiRNAs, microRNAs that have been associated with specific cancer forming events, has been indicated in tissue, blood, and urine samples. These dysregulated miRNAs have an influence on several biological elements and signalling pathways, including PI3K/Akt [71,91,92]. The involvement of miRNAs in bladder cancer disorders shows that miRNAs may be considered as potential therapeutic targets to contrast multidrug resistant situations [93].

Numerous studies have shown that changes in miR27a3p expression are associated with chemoresistance not only in bladder, gastric, and ovarian carcinoma but also in leukaemia, hepatocellular cancer, and breast cancer through the PI3K/Akt signalling pathway [94,95,96,97,98]. In particular, new evidence suggests that miR520h is responsible for breast cancer cell resistance to paclitaxel by attenuating the stability of the tumour suppressor PTEN and by activating the Akt pathway [99].

In the whole, these findings indicate a new starting point to overcome drug resistance based on strategies that target the implicated miRNAs.

**Table 1 cancers-13-03949-t001:** The regulation of miRNAs on the PI3K/Akt pathway throughout different protein targets.

Study	miRNA	Tumour Type	Regulation	Target	Effect
Xu X et al. [75]	miR-182-5p	RCC	Negative	AKT/FOXO3a via FLOT1	Cancer proliferation
Lian JH et al. [76]	miR-122		Positive	PI3K/AKT	Cancer proliferation
Huang J et al. [69]	miR-205-5p		Negative	PI3K/Akt/mTOR and VEGFA	Tumour suppression
Yang SM et al. [81]	miR-21	Gastric	Negative	PTEN/PI3K/Akt	Drug resistance
Li NA et al. [83]	miR-196b		Positive	PI3K/Akt/mTOR	Tumour promoting
Tsukamoto Y et al. [84]	miR-375		Negative	PKD1/Akt and 14-3-3zeta	Tumour suppression
Peng Y et al. [85]	miR-34		Negative	PDGFR/Akt and MET	Tumour suppressor
Wang T et al. [86]	miR15a, miR16		Negative	TWIST1	Tumour suppressor
Kang W et al. [87]				YAP1	
Wu L et al. [88]	miR137		Negative	AKT2	Tumour suppressor
Liang M et al. [89]	miR203		Negative	PIK3CA/Akt	Tumour suppressor
Riquelme I et al. [90]	miR125b-2, miR-451a and miR101-2		Negative	mTOR via PIK3CB and TSC1	Tumour suppressor
Deng Y et al. [91]	miR27a	Bladder	Negative	RUNX-1	Drug resistance
Zhao X et al. [94]		gastric,	Negative	p21	
Li Z et al. [95]		ovarian	Negative	HIPK2	
Feng DD et al. [96]		leukaemia	Positive	MDR1	
Chen Z et al. [97]		hepatic	Positive	FZD7/β-catenin	
Zhu B et al. [98]		breast	Positive	BTG2	
Geng W et al. [99]	miR-520h	Breast	Positive	OTUD3/PTEN	

RCC: renal cell carcinoma; FLOT1: flotillin-1; FOXO3a: (Forkhead Transcription Factor O Subfamily Member 3a); PTEN: phosphatase and tension homolog deleted on chromosome 10; PKD1: polycystic kidney disease 1; PDGFR: platelet-derived growth factor receptor; MET: tyrosine-protein kinase Met; TWIST1: oncoprotein with important roles in the epithelial to mesenchymal transition process; YAP1: Yes-associated protein 1; PIK3CA: enzymatic subunit p110α of phosphatidylinositol 3-kinase; TSC1: tuberous sclerosis complex 1; RUNX-1: runt-related transcription factor 1; HIPK2: homeodomain-interacting protein kinase-2; MDR1: multiple drug resistance protein 1 gene; FZD7: frizzled class receptor 7; BTG2: B-cell translocation gene 2.

## 5. Inhibition of the PI3K/Akt Pathway

The PI3K/Akt/mTOR pathway is activated by many factors targeting different members of the pathway [95]. The primary negative regulator of this pathway is the tumour suppressor PTEN; it has lipid and protein phosphatase activity by inhibiting PI3 phosphorylation and catalysing the reaction opposite to PIP3 generation, the conversion of PIP3 to PIP2 [100,101]. In fact, PIP3 levels are significantly increased in PTEN-knockout mice compared to wild mice, and Akt is constantly activated [102]. PTEN is able to interfere with the accumulation of Akt in the cell membrane and leads to the release of inactive Akt to the cytoplasm, thus acting downstream of the Akt pathway [103]. The phosphorylation of one of its phosphorylation sites at the c-terminal tail allows PTEN to attain a closed state, thereby determining the protein stability and consequent inactivation; on the contrary, its dephosphorylating contributes to an increase in activity [104]. Other phosphatases, apart from PTEN, are responsible for the termination of the transduction signalling, at downstream level of PI3Ks. Among them, the family of Src-homology 2-containing inositol 5′-phosphatase phosphatases (SHIP) plays a fundamental role in the negative regulation of Akt [105,106]. 

Many drugs, like PI3K inhibitors, targeting single or more proteins, include LY294002, which operates by inactivating PKB and determining cell cycle arrest. In addition, a new generation of PI3K inhibitors in preclinical studies may induce apoptosis and have so far been well tolerated in the early stages of clinical trials [107,108,109].

Among Akt inhibitors, more evidence has encouraged the clinical application of Palomid529 and perifosine, both of which enhance the radio-sensitivity of tumour cells [110,111]. Furthermore, mTOR is a sensitive target for rapamycin and seems to be the most qualified target belonging to PI3K/Akt pathway. Classically, rapamycin has been used as an immunosuppressant agent to prevent organ transplant rejection, but now it is also being employed as a tumour suppressor against mTOR [112,113]. Clinical results obtained from mTOR inhibitors support oncologists in actively developing these drugs [114]; a brand new rapamycin, everolimus, selectively inhibits mTORC1 and preserves mTORC2 [115]. New emerging therapeutic strategies based on dual PI3K/Akt/mTOR inhibitors are able to block mTOR as well as PI3K and Akt activation maintaining a good tolerance in the clinical trials. In this way, the pathway is inhibited at different layers in order to increase the efficiency of new combination therapies [116].

## 6. The PI3K/AKT Pathway and Drug Resistance

PI3K signalling and its downstream effecter Akt are considered significant causes of chemoresistance in cancer therapy in a variety of tumours [117,118,119,120,121,122]. Abnormal initiation of the pathway inhibits chemotherapeutic-induced apoptosis by multiple mechanisms, such as reinforcing the action of anti-apoptotic genes and counteracting the pro-apoptotic genes [123]. Bcl-2 proteins, located on the mitochondrial membrane, include anti-apoptotic proteins, such as BCL-xl and Bcl-2, and pro-apoptotic proteins, such as Bax and Bad [124,125]. Activated Akt stimulates PAK1, which, in turn, phosphorylates Bad, causing its release from the mitochondrial membrane to the cytoplasm and, at the same time, inhibits the translocation of Bax from the cytoplasm to the mitochondria. Therefore, the hyper-activation of Akt promotes cell survival via the stimulation of Bcl-2 and via the inhibition of Bax, which are in charge of cancer cell resistance.

Bcl-2 acts as an oncoprotein, and its overexpression correlates to multidrug resistance (MDR). Since cancer cells exploit Bcl-2 to evade apoptosis, it appears that in many tumours there is a selection of Bcl-2-dependent cells [126,127]. Although the overexpression of Bcl-2 predisposes to tumour invasiveness, some studies have reported that it may represent a crucial target for Bcl-2 inhibitors and, therefore, a good prognostic factor in patients with different types of cancer [128,129,130].

An abnormal change in the pathway also induces the upregulation of X-linked inhibitor of apoptosis (Xiap), a downstream effector of Akt which prevents apoptosis in several ways, one of which is the inhibition of autophagy via regulation of the cytosolic level of p53 [131]. Normally, phosphor-Xiap interacts with Mdm2 and mediates its rapid degradation, thereby maintaining the relatively high levels of cytosolic p53. Under stress conditions, the Akt inhibition determines the dephosphorylation of Xiap, thus promoting dissociation from Mdm2. This makes Mdm2 stable and, as a consequence, leads to the fast degradation of p53, facilitating at the same time the induction of autophagy [131].

XIAP belongs to the IAP family and plays a crucial role in inhibiting caspase-3, caspase-7, or caspase-9 [75]. IAPs were found to be over-expressed in many cancer types and to be related to tumour cell survival, leading to poor prognosis and overall survival [132,133,134,135,136,137].

For this reason, the use of Xiap inhibitors, alone or in combination with PI3-kinase inhibitors, is considered to be a potential therapeutic strategy for the management of cancer [137,138,139].

The inhibition of apoptosis and the advancement of tumour growth may be due to the over-activation of the NF-kB system generated by the triggering of PI3K/Akt [140,141]; as result, NF-kB is able to accelerate the cell cycle process in tumour cells, contributing to the development of multidrug-resistance [142,143].

The activation of PI3K/Akt signalling inactivates FOX factors, which are involved in numerous physiological processes, including cell division, cell death, cell invasion, and drug resistance [144].

It has been reported that the over-expression of Akt1 by the activation of HER2/PI-3K confers a broad-spectrum chemoresistance on breast cancer cells [145]. Likewise, the hyper-expression of paired-related nomeobox-1 (PRRX1) determined by the activation of PTEN/PI3K/AKT signalling leads to the epithelial-to-mesenchymal transition (EMT) implicated in multiple processes, such as cell invasion and MDR in breast cancer [146].

In addition, as explained above, the dysregulation of micro-RNA that exerts effects on the PI3K pathway may also exacerbate multidrug resistance in cancer cells. These miRNAs could regulate MDR through targeting a specific gene or targeting several genes simultaneously, including genes affecting the response of cells to chemotherapy drugs. One miRNA can target different messenger RNAs (mRNAs), and, on the other hand, one mRNA can be regulated by many miRNAs. Given the complexity and heterogeneity of tumour cells, the effects exerted by miRNAs on drug resistance depends on the different kinds of tumours and differ from tumour to tumour [147].

## 7. Target Therapies and Future Prospective

In the previous sections, we have analysed the principal alterations occurring in the PI3K/Akt pathway in human malignancies. Given that this cascade is considered the most frequently activated pathway in standard cancers, its members represent future targets for anti-cancer drugs, including nearly isoform-specific PI3K inhibitors, dual inhibitors, mTOR and Akt inhibitors, and miRNA inhibitors [148].

In recent years, targeted drugs for each molecular site and domain involved in different tumours have gained further development and validation. Thus, it is reasonable to think that with the continuous exploration of the underlying mechanisms, the discovery of more promising drug targets is becoming the best therapeutic approach for cancer treatment and prevention. With the purpose of enhancing the sensitivity of tumour cells, PI3K/Akt inhibitors have been recently tested with success [149]. For instance, in breast cancer, the introduction of PI3K/Akt inhibitors such as a novel drug, econazole, has been able to counter the activation of PI3K/Akt pathway and, in turn, to prevent the resistance of tumour cells by exerting cytotoxic activity [150]; the treatment with phenybutyl isoselenocyanate (ISC-4) considerably slows down the activation of PI3K/Akt in a dose-dependent way [151]. Consequently, the use of specific inhibitors of this signalling pathway, alone or in combination, is currently under evaluation, for the development of brand new strategies in order to counter the drug resistance of various cancers.

In many cases, the therapeutic resistance emerges as a consequence of several mechanisms that contrast the suppression of the signalling pathway and cause, as a form of compensation, the activation of other signalling pathways. Thus, clinical or preclinical trials are starting to determine the efficiency of various combination therapies to address different levels of inhibitors pathways. 

As previously mentioned, IAPs are emerging as promising targets for cancer therapy, and several clinical trials based on IAPs antagonist are currently under evaluation [152].

The implication of changes in the miRNA expression profile may lead to an increase of PI3K/Akt/mTORC1 activity that perpetuates and exacerbates the multidrug resistance in cancer. The miRNAs and their intricate networks have arisen as fundamental biological elements with a high potential for clinical application. As a matter of fact, different strategies to improve miRNAs delivery have been performed on the basis of their modulation in cancer cells [153]. Although it is expected that specific inhibitors of miRNAs can reduce the multidrug resistance of cancers via deregulation of the PI3K/Akt pathway, at present, no therapy targeting specific miRNAs has been performed in medical practice, and further studies are required.

## 8. Conclusions

Significant progress has been recently made in the management of cancer drug-resistance. The new challenge is to overcome the limitations of the classical therapeutic strategies, which may be met with the introduction of novel agents inhibiting not only PI3K/Akt targets but other members of the same pathway. The correct combination of available drugs and the discovery of new potential biomarkers, such as miRNAs, are needed and warranted.

## Figures and Tables

**Figure 1 cancers-13-03949-f001:**
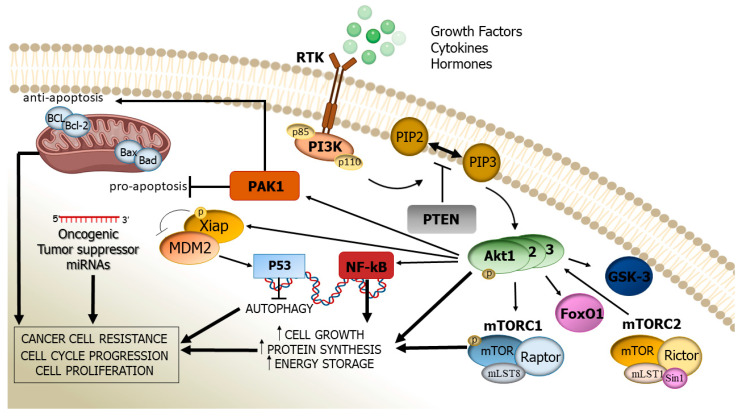
Involvement of PI3K/Akt/mTOR signalling pathways in the development of cancers. As illustrated, the auto-phosphorylation of PI3K leads to the activation of Akt, through PIP2 to PIP3 conversion. PI3K/Akt and mTORC1 contribute to tumour growth and the energy storage of cancer cells. The upregulation of Akt Promotes the phosphorylation of target genes and proteins which promote the inhibition of apoptosis and autophagy. Deregulated miRNAs also contribute to cancer cell proliferation.

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
