# Peer review of "The Pathogenic Role of PI3K/AKT Pathway in Cancer Onset and Drug Resistance: An Updated Review"

_cancers, 2021, doi:10.3390/cancers13163949_

Round 1

Reviewer 1 Report

The authors have responsed by concerns.

Reviewer 2 Report

I think this paper is fit for publication.

Reviewer 3 Report

The revised manuscript is recommended to be accepted in the present form.

Reviewer 4 Report

While the authors improved the manuscript, the lack of originality over the broad variety of existing reviews and many conceptual mistakes persist.

E.G The statement “Generally speaking, PI3K is activated by extracellular signals, such as growth factors, cytokines and hormones, linking the N-terminal extracellular domain of the transmembrane receptor tyrosine kinase (RTK)” is misleading.

Cytokines and most hormones usually don´t signal through RTKs. In addition, the authors omit the activation of PI3K through Ras. The statement “There are several Akt target proteins, such as FoxO1, GSK-3, mTOR” is a very particular selection as FOXO1, 3 and 4 are AKT substrates and mTOR does not present a direct AKT substrate. mTOR activation by AKT is mostly indirect. Therefore, this does not provide a mechanistic understanding of the pathway. The pathway has several negative feedback loops which are of importance therapeutically. The statement “In addition, lipid and protein phosphatases controlling Akt activity are frequently downregulated in human cancer determining the overexpression of phosphorylated Akt and the subsequent increase of PI3K/Akt pathway activity” without naming the responsible enzymes is yet another misleading understatement taking into account that the phosphatase that antagonizes PI3K, PTEN is one of the most important tumor suppressors and frequently lost in cancer.

This manuscript is a resubmission of an earlier submission. The following is a list of the peer review reports and author responses from that submission.

Round 1

Reviewer 1 Report

The current manuscript Rascio et al entitled “The pathogenic role of PI3K/AKT Pathway in cancer onset and drug resistance: un updated review” timely reviews several aspects of the PI3K/AKT Pathway in cancer biology. However, the manuscript is not always well written, is weirdly referenced and lacks originality over the broad variety of existing reviews. Most importantly, the manuscript contains conceptual errors and inaccurate statements, in particular within the “Structure, activation and function of PI3K/Akt cascade” chapter and therefore does not promote a mechanistic understanding of the pathway.

Author Response

We modified our manuscript, we corrected the spelling errors, we added new references and revised all the review

Reviewer 2 Report

The manuscript entitled "The pathogenic role of PI3K/AKT Pathway in cancer onset and drug resistance: un updated review" gave a very concise review of this very significant oncogenic pathway  and its application as therapeutic target in the development of anti-cancer drugs.

Major comments:

  1. The manuscript is well-written and well-organized, and it summarizes the fundamental factors in this pathway.
  2. However, there wasn't much description of the downstream effectors of this pathway (e.g. S6K, 4E-BP1...) neither their roles in the research in the anti-cancer drugs. 

Author Response

We revised the manuscript, as seggested

Reviewer 3 Report

The paper is conceptually very appealing. The authors argue that PI3K/AKT pathway is one of the most frequently over-activated intracellular pathway in several human cancers. This pathway, acting on different downstream target proteins, contributes to the carcinogenesis, proliferation, invasion, metastasis of tumour cells.

In this study, the authors investigated he involvement of this pathway in patho-physiological cell survival mechanisms, emphasizing its key role in the development of drug-resistance.

PI3K inhibitor and Akt inhibitor can be elaborated adding few more references highlight the importance in cancer onset and drug resistance. 

I suggest authors to thoroughly check minor english mistakes and few changes to sentences.

Author Response

we revised the manuscript. Thanks for the suggestions

Reviewer 4 Report

In the current manuscript, the authors underlined the involvement of PI3K/AKT pathway in patho-physiological cell survival mechanisms, emphasized its key roles in the development of drug-resistance. They also provided an overview on the potential inhibition strategies currently available. Overall, this review summarized some progresses for the field, but did not describe much detail on drug resistance as the title mentioned, and also the references have some certain of limitations, like low-impact and outdated.

Author Response

We expanded our references regarding the drug resistance. Thank you for your suggestions